# Beyond Astana: Configuring the World Health Organization Collaborating Centres for primary health care

**Resham B. Khatri**[1,2]*, **Peter S. Hill**[1], **Eskinder Wolka**[3], **Frehiwot Nigatu**[3], **Anteneh Zewdie**[3], **Yibeltal Assefa**[1]

**1** School of Public Health, the University of Queensland, Brisbane, Australia, **2** Health Social Science and Development Research Institute, Kathmandu, Nepal, **3** International Institute for Primary Health Care-Ethiopia, Addis Ababa, Ethiopia

* rkchettri@gmail.com

**Data Availability Statement:** All relevant data are included within the manuscript.

**Funding:** The author(s) received no specific funding for this work.

## Abstract

The understanding of primary health care (PHC) has evolved significantly, evident in key World Health Organization (WHO) reports, promoting PHC as a means for health for all, identifying key health systems reforms and focusing on health care experience. This study explores the WHO's current framing of PHC, and its configuration of WHO Collaborating Centres (WHOCCs) on PHC using the data available on the WHOCCs Portal. We analysed the following variables: title, institutions, location, economy, date of mandate, objectives, subject, and activity. There were 13 WHOCCs on PHC, nine based in North America and Europe, and none in Africa. Only three were in Low- and Middle-Income Countries (LMICs). The WHOCCs on PHC focused on three broad subjects: five focused on human resources for health (HRH); four on health systems research (HSR) and development, with an emphasis on family medicine; four on PHC systems. Activities were related to training and education, provision of technical advice, and research. Support to WHO on implementation of PHC was an activity for two LMIC based WHOCCs. The current configuration of WHOCCs on PHC is consistent with the evolution of PHC and its intersection with Universal Health Coverage and the Sustainable Development Goals. The increasing attention to people-centred health systems aligns with WHO's commitment to PHC in all health systems, though this needs special interpretation for LMICs with their limited HRH. There has been a shift in subjects from HRH towards primary care and family medicine, and HSR highlighting primary care and PHC systems. The concern is an absence of WHOCCs in the Africa and Latin and South Americas, and under-representation in LMICs. Designating more institutions from the South with expertise in PHC is necessary to address the challenges post-Astana.

## Introduction

In 1978, the Alma-Ata Declaration of Primary Health Care (PHC) enunciated principles of comprehensiveness, intersectoral coordination, the use of appropriate technology, and affordable and appropriate health services for all [1]. It broadened the perception of health beyond a

**Competing interests:** The authors have declared that no competing interests exist.

merely sectoral approach, redefining health as a human right. Over the last four decades, PHC has been reiterated and progressively reformulated (Fig 1). Four decades later, in 2018, *A Vision for Primary Health Care in the 21st Century*) [2] provided the rationale for, and the foundation of the Declaration of Astana on PHC [3]. It identified five WHO reports as key to the ongoing evolution of PHC: Global Strategy for Health for all (HFA) by the Year 2000 [1]; Primary Health Care 21: "Everybody's Business" [4]; Final report of the Commission on the Social Determinants of Health (SDH) (2008) [5]; World Health Report (WHR) (2008): Primary Health Care–now more than ever [6], and Framework On Integrated, People-centred Health Services (IPCHS) 2016 [7].

## Inception, reaffirmation, and broadening scope of PHC

The Alma-Ata declaration was integral to the global strategy for HFA by 2000. The PHC strategy elaborated the extension of health care coverage in keeping with the basic right of all humans to enjoy the many benefits of health, economic productivity and social activity-intersectoral activities that reinforce one another and contribute to human and health development [1]. It emphasized delivery of country-wide programs that would reach the whole population through medical interventions and appropriate technology, that are scientifically sound, adaptable to local context, acceptable to the community, with active community participation, and affordable within country resources. The PHC strategy spelled out the need for international action to support these initiatives through information exchange, promoting research and development, technical support, training, and ensuring coordination within and beyond the health sector [1].

The 2000 report "Primary Health Care 21: Everybody's Business" further reaffirmed the relevancy of the Alma-Ata vision [4]. The report expressed the need for revitalisation of PHC to operationalise the values of Alma-Ata, by developing sustainable health systems and establishing complementary systems for governance. It reiterated the advancement of PHC through adequate capacity (e.g., infrastructure, technical expertise and political will) for the revitalisation of the commitment to PHC (e.g., building on partnerships, empowerment of communities, and commitment of public and private sectors and international organizations) [4].

While the Declaration on PHC was grounded in health rights, and sought to address inequity, in 2008, the Final Report of the Commission on the Social Determinants of Health (SDH) demonstrated how structural and intermediary determinants of health contributed to those

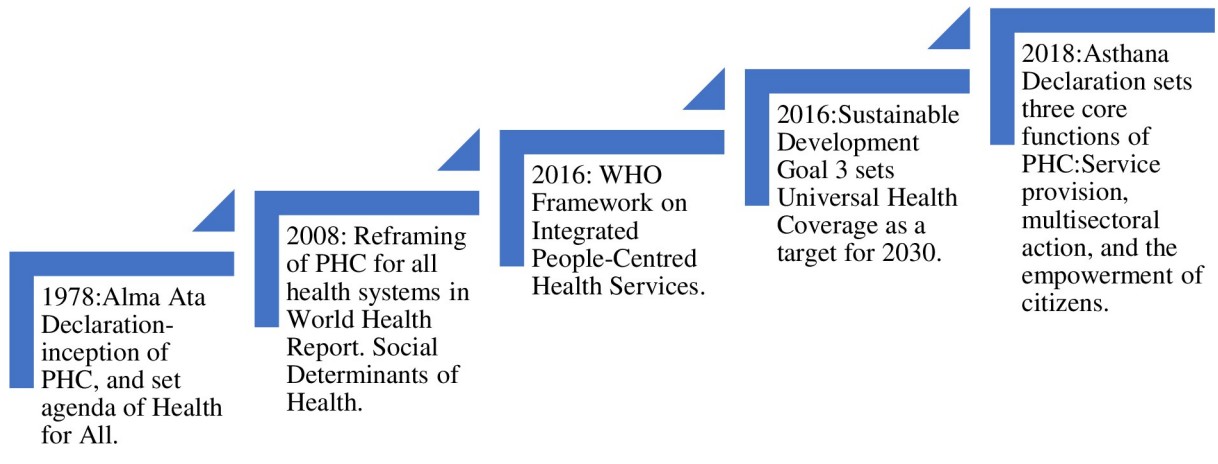

**Fig 1. Summary of development and revolution of PHC since 1978.**

inequities. Underlining the intersectoral contributions to health, the report proposed three principles of action to close the gap in health inequities:

- improving the conditions of daily life (the circumstances in which people are born, grow, live, work, and age);

- tackling the inequitable distribution of power, money, and resources (e.g., the structural drivers of those conditions of daily life); and

- measuring the problems, evaluating actions, expanding the knowledge base, developing a workforce and raising public awareness about the social determinants of health [5].

## Shift toward primary care

The WHR 2008: Primary Health Care- now more than ever, set itself two tasks: to reaffirm the importance of PHC, and to reframe it for a global environment, relevant to states at all stages of development. It articulated four sets of reforms required to translate PHC values into policy: universal coverage reforms to engage the whole population; service delivery reforms reorganised around primary care, with a focus on teams of health workers facilitating access to care, and the appropriate use of modern technology and medicines [6]; public policy reforms that integrated public health initiatives into primary care; and leadership reforms that adopted a participatory approach to policy implementation. In addition, the role of the hospital was reframed in support of primary care networks, supporting them in coordinating a comprehensive response at all levels [6].

That shift in focus toward primary care was further reiterated in the WHO Framework on Integrated People-Centred Health Services in 2016 [7]. The IPCHS framework emphasises building strong primary care-based systems for reaching the entire population and guaranteeing universal access to services. To achieve this, IPCHS proposes two policy interventions: a) increasing primary care services (family medicine, multidisciplinary PC teams and health expenditure allocated to primary care); and b) shifting towards more outpatient care (ambulatory care, home care, nursing homes and hospices, repurposing hospitals for acute care, surgery and patient care) [7].

## Redefining of PHC for the 21st century

The WHO's redefined vision of PHC for the 21st century frames PHC as a whole-of-society approach to health to ensure equity in the highest possible level of health and wellbeing. This vision emphasises integrated services (primary care and essential public health) to address people´s needs and preferences (as individuals, families, and communities). It seeks engagement as early as possible along the continuum from health promotion and disease prevention to treatment, rehabilitation, and palliative care, with access as close as feasible to people. Integrated services comprise both whole of person-centred care (first contact, comprehensive, continuity, person-centred) and population-based services (health protection, health promotion, disease prevention, surveillance and response, and emergency preparedness) [2].

The health-related Sustainable Development Goal (SDG) 3 seeks to "ensure healthy lives and promote well-being for all at all ages" by 2030. The goal picks up and makes central one of the key WHR 2008 PHC reforms: universal health coverage (UHC) with its access to good-quality health services for all, without financial hardship [2]. That conjunction of PHC and UHC was reflected in the 2018 Declaration of Astana, where governments reaffirmed their commitment to the SDGs, redefining the three core functions of primary health care: service provision, multisectoral action and the empowerment of citizens [3]. These current global

policies, activities and actions have shaped the Special Programs on PHC, and are reflected in the development of the WHO Collaborating Centres (WHOCCs) on PHC over the past three decades.

The WHOCC is an institution designated by the Director-General of WHO to form part of an international collaborative network set up by WHO to support its programme at the country, intercountry, regional, interregional and global levels. In line with the WHO policy and strategy of technical cooperation. A WHOCC also contributes to strengthening country resources, in terms of information, services, research and training in support of national health development (WHO). Currently, more than 800 WHOCCs in over 80 member states are working with the WHO on areas (nursing, occupation health, communicable disease, nutrition, mental health, chronic diseases and health technologies) [8].

This study examines how WHO currently configures its WHOCCs on PHC, their distribution, focus subjects, activities and objectives, as PHC has been progressively redefined. It seeks to understand how these WHOCCs complement the evolution of PHC in the 21st century, and what is needed to shape its future development.

## Methods

We identified the WHOCCs on PHC, analysing their available profile information as listed on the WHO Collaborating Centres Database and Portal (https://apps.who.int/whocc/Default.aspx). In the Database, there were following information of each WHOCC: title, institution, geographic location (address, town country, and region), dates (designation, redesignation and expiry), objectives, focus subjects, types of activity, and WHO outputs. We focussed on the analysis variables: title, kinds of institutions, location (country, region), economy, and date of mandate. We summarised background information of WHOCCs using title, institutions, dates, and geographic variables. Second, we categorised WHOCCs into three broader categories using their titles (health workforces; family medicine and primary care; and PHC and PHC systems). Using these broader categories, we analysed the objectives and focused subject analysis, objectives and activities of CCs.

## Findings

### Description of WHOCCs

There are 13 WHOCCs on PHC, 12 active and one pending, identified by the Special Program for PHC. Of these, ten are based in high-income countries (HICs); two are in upper-middle-income countries (UMICs) (Shanghai, China; Moscow, Russian Federation), and one is in a low-income country (Pyongyang, Democratic People's Republic of Korea). Five are in the European Region (EURO), four in the Americas Region (AMRO), and three in the Western Pacific Region (WPRO). One WHOCC for PHC Development (Pyongyang, Democratic People's Republic of Korea) based in the Southeast Asia Region (SEARO) is pending. Ten are departments of academic institutions. The three WHOCCs affiliated with Ministries of Health are with state-controlled regimes; particularly two from Upper-Middle-Income Countries (UMICs) (People's Republic of China, and the Russian Federation); and one from Low-Income Country (Democratic People's Republic of Korea) (see Table 1).

The first wave of WHOCCs, established post- Alma Ata declaration (designated between 1986–92) had human resources for health as their in their titles. The second wave of WHOCCs on PHC (2013–20) which were established mostly pre-Astana Declaration designated their titles around PHC, primary care, family medicine and health system development with targeting specific challenges such as NCDs, patient safety and mental health.

**Table 1. Descriptive characteristics of WHOCCs on PHC by WHO Region as of 2022.**

| | WHOCC Title | Institution | Designation | Redesignation | Expiry | Location | Economy | Region |
|---|---|---|---|---|---|---|---|---|
| 1 | WHOCC for Quality and Equity in PHC Systems | Academic | Mar 2017 | Mar 2021 | Mar 2025 | Amsterdam, Netherlands | HIC | EURO |
| 2 | WHOCC on PHC | Academic | Oct 2013 | Oct 2021 | Oct 2025 | Granada, Spain | HIC | EURO |
| 3 | WHOCC for the evidence-based management and prevention of NCDs in primary care | Academic | Jul 2019 | Jul 2019 | Jul 2023 | Oxford, UK | HIC | EURO |
| 4 | WHOCC on Family Medicine and PHC | Academic | Apr 2020 | Apr 2020 | Apr 2024 | Ghent, Belgium | HIC | EURO |
| 5 | WHOCC on primary care competence in mental health and psychiatric crisis interventions in the community | Ministry of Health | Feb 2013 | Feb 2021 | Feb 2025 | Moscow, Russian Federation | UMIC | EURO |
| 6 | WHOCC for International Nursing Development in PHC | Academic | Jun 1986 | Apr 2019 | Apr 2023 | Chicago, USA | HIC | AMRO |
| 7 | WHOCC in Primary Care Nursing and Health Human Resources | Academic | Nov 1992 | Apr 2019 | Apr 2023 | Hamilton, Canada | HIC | AMRO |
| 8 | WHOCC on Family Medicine and Primary Care | Academic | Sep 2018 | Sep 2018 | Sep 2022 | Toronto, Canada | HIC | AMRO |
| 9 | WHOCC for Medical Education and PHC | Academic | Aug 2014 | Aug 2018 | Aug 2022 | Rockford, USA | HIC | AMRO |
| 10 | WHOCC for PHC | Government Unit | Jun 2014 | Jun 2020 | Jun 2024 | Shanghai, China | UMIC | WPRO |
| 11 | WHOCC for Research and Training for Nursing Development in PHC | Academic | Jan 1988 | Mar 2020 | Mar 2024 | Seoul, Republic of Korea | HIC | WPRO |
| 12 | WHOCC for Nursing Development in PHC | Academic | May 1990 | Apr 2020 | Apr 2024 | Tokyo, Japan | HIC | WPRO |
| 13 | WHOCC for PHC Development | Ministry of Health | Aug 1988 | Jul 2018 | Jul 2022 | Pyongyang, DPRK | LIC | SEARO |

PHC: Primary Health Care. WHOCC: World Health Organization Collaborating Centre, EURO: European Regional Office, SEARO: Southeast Asia Regional Office, WPRO: Western Pacific Regional Office. HIC: High-Income Country, UMIC: Upper-Middle-Income Country, LIC: Low-Income Country. DPRK: Democratic People's Republic of Korea

Seven WHOCCs mention PHC in their title, and six WHOCCs have titles that refer to primary care or family medicine. Ten of the thirteen WHOCCs have renewed their mandates since the Astana Declaration (October 2018). All current WHOCCs have mandates that will be due for renewal by 2025.

## Analysis of subjects and objectives for WHOCCs on PHC

The subject descriptors identified for WHOCCs include Human Resources, Research and Policy Development, Health Systems Research (HSR) & Development, Health Promotion & Education (HPE), Social Determinants of Health (SDH), Patient Safety, and Non-Communicable Diseases (mental health, and other non-specified). The health workforce and research were heavily represented: eight WHOCCs focus on HRH (four each for nursing and HRH other than nursing), while eight WHOCCs focus on research and development (six on HSR & Development, and two on Research Policy & Development), and four focus on HPE. A range of other specific subjects are covered in individual WHOCCs: Patient Safety, Mental Health and Neurosciences, SDH, and NCDs (Table 2).

The objectives broadly correspond with the subject areas. For example, Table 2 shows five WHOCCs with subjects and objectives that focus on HRH (Yellow), and four CCs that focus on family medicine, primary care, and NCDs (Pink). Four CCs have subjects and objectives that focus on PHC Health Systems Research and development (Blue). However, not all subjects

**Table 2. WHOCCs on primary health care and focus subjects, as of 2022.**

| WHOCCs | City, country | HRH (Nursing) | HRH (other than nursing) | HSR & development | Research policy & development | Health promotion & education | Patient safety | Mental health & neurosciences | SDH | NCDs other than specified |
|---|---|---|---|---|---|---|---|---|---|---|
| WHOCC for Nursing Development in PHC | Tokyo, Japan | Yes | | | | Yes | Yes | | | |
| WHOCC for Medical Education and PHC | Rockford, USA | | Yes | | | Yes | | | Yes | |
| WHOCC for International Nursing Development in PHC | Chicago, USA | Yes | Yes | | | | | | | |
| WHOCC for Research and Training for Nursing Development in PHC | Seoul, South Korea | Yes | | | | Yes | | | | |
| WHOCC in Primary Care Nursing and Health Human Resources | Hamilton, Canada | Yes | Yes | | | | | | | |
| WHOCC on Family Medicine and Primary Care | Toronto, Canada | | | Yes | | | | | | |
| WHOCC for the evidence-based management and prevention of NCDs in primary care | Oxford, UK | | | Yes | | | | | | Yes |
| WHOCC on Family Medicine and PHC | Ghent, Belgium | | | Yes | | | | | | |
| WHOCC on primary care competence in mental health and psychiatric crisis interventions in the community | Moscow, Russian Federation | | | | | | | Yes | | |
| WHOCC on PHC | Granada, Spain | | | Yes | | | | | | |
| WHOCC for PHC | Shanghai, China | | | Yes | | | | | | |
| WHOCC for Quality and Equity in PHC Systems | Amsterdam, Netherlands | | | Yes | Yes | | | | | |
| WHOCC for PHC Development | Pyongyang, DPRK | | Yes | | Yes | Yes | | | | |

Yellow: WHOCCs with priority HRH (nursing), health promotion and education, and patient safety.

Pink: WHOCCs with priority on family medicine, primary care, and NCDs-related health systems research & development, and mental health and other NCDs.

Blue: WHOCCs related to PHC systems, equity, and quality with a focus on health systems research & development; and research policy & development.

HSR: Health Service Research. SDH: Social Determinants of Health. HRH: Human Resources for Health. DPRK: Democratic People's Republic of Korea.

are linked to related objectives: only one of four WHOCCs with HPE as a subject has an objective specifically relating to this area, and only two WHOCCs of the four nominated as dealing with HRH other than nursing have a specific related objective. Similarly, Patient Safety and SDH are each listed as subjects, but no corresponding objective has been stated.

## Human resources for health

Five WHOCCs have a health workforce focus, covering five subject areas: HRH nursing) (4), HRH (other than nursing) (3), health promotion & education (3), patient safety (1), and SDH (1) (Yellow). Three WHOCCs (Chicago, Tokyo, and Seoul) focus on capacity development and nursing education for PHC. The WHOCC in Hamilton focuses on developing and disseminating tools on best practices of nursing leadership and practices. The WHOCC (Seoul) focuses on generating and disseminating evidence on nursing workforce models for improved PHC services. The WHOCC (Hamilton) supports HRH assessment, planning, and data collection to inform education research and practices. The WHOCC in Rockford offers support to conceptualise, implement and evaluate education to prepare future health care providers in PHC, and supports WHO in reorientating medical education towards accountability, and developing education programs to serve underserved communities.

The WHOCC (Tokyo) focuses on health promotion & education, including development of people-centred models of care in the context of ageing societies. In addition, this WHOCC focuses on documenting and sharing lessons on implementing health literacy programs and engagement of communities and households with health care providers. Two WHOCCs (Rockford and Seoul) have HPE as their focus subject, with Tokyo focusing on patient safety and Rockford on SDH.

## Family medicine, primary care, and NCDs

Four WHOCCs focus on family medicine, primary care as part of HSR and development, and NCDs (Pink). Objectives under HSR and Development include support in the development of technical products (e.g., standards), research and evidence generation on PHC and people-centred care to improve quality and equity, promotion of the development of PHC and primary care-based systems, training of family physicians, and NCDs management in PHC. Two WHOCCs (Shanghai and Toronto) describe their focus as people-centred integrated care, while two WHOCCs (Toronto and Ghent) focus on family medicine and primary care, including the integration of primary care into training programs. The Oxford WHOCC supports WHO on the management of NCDs in primary care settings.

The Moscow WHOCC focuses on mental health and neurosciences with specific support to WHO's work in advancing evidence-based suicide prevention programs. In addition, the Oxford WHOCC focuses on NCDs supporting WHO in the development of evidence-based technical documents to reduce premature mortality from NCDs.

## PHC health systems research and development

Four WHOCCs specifically address PHC, quality and equity in PHC systems, and PHC development (Blue). Three WHOCCs (Granada, Ghent, and Shanghai) focus on HSR and development, with the Granada WHOCC focusing on PHC-based health systems. The Amsterdam WHOCC focuses on research and technical assistance for improved quality and equity of PHC systems. The Shanghai WHOCC supports collaborative research on service delivery and training models to expand access to PHC and essential public health.

Similarly, two WHOCCs (Pyongyang, and Amsterdam) have Research Policy & Development objectives. The Pyongyang WHOCC (currently pending) focuses on researching and implementing actions to improve access to PHC service packages. This WHOCC also supports knowledge generation and strengthening professional networks and exchange practices in PHC, focusing on integrated people-centred health services. The WHOCC for Quality and Equity in PHC Systems (Amsterdam) aims to support member-states to advance the actionable measurement of PHC performance for quality and equity-oriented strategies.

## Analysis of the activities of the WHOCCs on PHC

The objectives defined under the subjects are further unpacked in a series of activities to be undertaken by the WHOCCs, which then lead to defined outputs. However, the activities and outputs do not appear to be directly linked to the objectives, but relate more broadly to the combined subjects and objectives.

Eight categories of activities are outlined (see Table 3): a) Training & Education; b) Information dissemination; c) Providing Technical Advice to WHO; d) Product Development (e.g., guidelines, standards); e) Research; f) Collection & Collation of Information, and g) Support

**Table 3. WHOCCs and outlined activities.**

| WHOCCs | City, country | Training and education | Technical advice to WHO | Information dissemination | Product development (e.g., guidelines) | Research | Collection & collation of information | Support to WHO in the implementation of programmes at country level |
|---|---|---|---|---|---|---|---|---|
| WHOCC for Nursing Development in PHC | Tokyo, Japan | Yes | | | | Yes | | |
| WHOCC for Medical Education and PHC | Rockford, USA | Yes | Yes | Yes | | | | |
| WHOCC for International Nursing Development in PHC | Chicago, USA | Yes | | Yes | | Yes | | |
| WHOCC for Research and Training for Nursing Development in PHC | Seoul, South Korea | | | | Yes | Yes | Yes | |
| WHOCC in Primary Care Nursing and Health Human Resources | Hamilton, Canada | Yes | | Yes | Yes | | | |
| WHOCC on Family Medicine and Primary Care | Toronto, Canada | Yes | Yes | | | | | |
| WHOCC for the evidence-based management and prevention of NCDs in primary care | Oxford, UK | Yes | Yes | | | | | |
| WHOCC on Family Medicine and PHC | Ghent, Belgium | | Yes | | | | Yes | |
| WHOCC on primary care competence in mental health and psychiatric crisis interventions in the community | Moscow, Russian Federation | | | | Yes | | Yes | Yes |
| WHOCC on PHC | Granada, Spain | Yes | | | | Yes | Yes | |
| WHOCC for PHC | Shanghai, China | | | Yes | | Yes | | |
| WHOCC for Quality and Equity in PHC Systems | Amsterdam, Netherlands | | Yes | | | Yes | | Yes |
| WHOCC for PHC Development | Pyongyang, DPRK | Yes | | | | Yes | Yes | |

Yellow: WHOCCs related to nursing and PHC-focused activities on technical support and research activities.

Pink: WHOCCs related to primary care, family medicine and NCDs, focusing on technical support and research-related activities.

Blue: WHOCCs related to PHC development and systems with a focus on technical support and research activities.

DPRK: Democratic People's Republic of Korea.

to WHO in the implementation of WHO programs at the country level. Broadly, the first four categories of activities can be grouped into i) technical support (training, education, information dissemination), ii) research (research, and collection and collation of information, and iii) implementation support to WHO programmes.

The categories of activities (Table 3) correspond broadly to the clusters for focus subjects and objectives listed in Table 2. For example, WHOCCs related to HRH focus on technical support (e.g., Training & Education, Information Dissemination, and Product Development) and activities related to research (Yellow). Similarly, WHOCCs related to family medicine, primary care, and NCDs focused activities were on providing technical advice to WHO, followed by Training & Education, and Collection and Collation of Information (Pink). The CC in Moscow focuses activities on Product Development, Implementation Support to WHO at the country level regarding mental health and psychiatric problems. Activities of WHOCCs related to primary health and PHC systems were on Research followed by technical support especially Training & Education, and Collection & Collation of Information (Blue). Support to WHO in the implementation of programmes at country level was listed as an activity in only two WHOCCs.

## Discussion

The five key WHO reports leading to the Astana Declaration outline a clear conceptual development of PHC, reiterating its grounding in the right to health for all [1], but evolving with social and economic improvement and the strengthening of health systems [4]. The Commission on the Social Determinants of Health [5] provided a detailed exploration of inequity and its health in the same year that the WHR 2008 [6] pointed to the relevance of PHC [5] to all health systems–detailing four sets of reforms. Universal Health Coverage–one of those reforms–was to become a target as the SDGs translated the multisectoral vision of PHC into sustainable development [9]. With PHC now claimed as the optimal approach to UHC [10], the focus has shifted to people-centred health care [7], with family medicine as a primary expression in mid- to high-income countries [11].

The configuration of WHOCCs on PHC reflects this transition. The first wave of WHOCCs, designated between 1986–92 (Chicago, Seoul, Pyongyang, Tokyo, Hamilton), had human resources for health as their subject focus. In four WHOCCs this focus was nursing, with training and education a key activity, and nursing specifically mentioned in their designation titles. Three of this first wave WHOCCs also focused on human resources other than nursing.

The second wave of WHOCCs on PHC (2013–20), designated around the introduction of UHC under the SDGs, shows a shift in focus towards health systems research and development, and targets specific challenges such as NCDs, patient safety and mental health. Only one WHOCC (Rockford) in this second round has its subject focus on human resources for health–in this case, doctors. Within this shift towards health systems focus, primary care and family medicine are prominent in the designation titles of centres designated after the Astana Declaration—Toronto and Ghent, with their focus on family medicine and PHC/PC, and Oxford, with its commitment to managing and preventing NCDs through PC.

The activities follow a similar trend: integral to this second wave of WHOCCs is a cluster of four centres linked specifically to PHC or PHC systems. Their focus is in PHC health systems research, and research policy and development, with activities that correlate with these areas. The Amsterdam WHOCC for Quality and Equity in PHC Systems specifies both provision of technical advice to WHO, and support in the implementation of programmes at country level.

Four of the five WHOCCs whose designation title includes primary care, family medicine or medical education, list technical advice to the WHO as a key activity, reflecting WHO's commitment to family medicine as an expression of people-centred health care. This shift towards people-centred care is consistent with the reforms on WHR 2008 [6], the IPCHS framework [7], and the United Nations recommendation (2010) on primary care, delivering "prioritised packages of essential interventions", supporting PHC, and empowering people for self-care [7]. The Astana declaration on PHC reiterates and endorses three principles—community empowerment, multisectoral policies and actions, integrated delivery of quality primary care and public health services—prioritising public health functions driven by knowledge and capacity building, human resources for health, technology, and financing [3, 12].

In summary, the progressive configuration of support to WHO through the WHOCCs on PHC reflects the evolving definition and positioning of PHC. The early focus on human resources for health has seen increasing attention to the medical workforce, consistent with the recognition of UHC as a global goal and PHC as a vehicle to achieve it. The socio-economic rise of many Low-and Middle-Income Countries (LMICs) has seen increasing disposable income, and increasing demand for medical care [13]. The introduction of technical advice to WHO as a listed activity among five second wave WHOCCs, suggests a conscious desire for feedback on this shift. At the same time, support on implementation has been added as an activity in the case of mental health programmes (Moscow) and PHC systems (Amsterdam). That interface between health systems research and development and support for the implementation of PHC programmes will be critical–particularly in LMICs–for the progressive achievement of UHC by 2030.

The current distribution of WHOCCs may be a challenge to achieving support for implementation: there are no WHOCCs in the African Region; no WHOCCs in the Pan-American Region come from Latin, Central or South America; the only WHOCC in the South-East Asian Region is pending; the three WHOCCs in the Western-Pacific Region are in high income countries, with none representing the Pacific. Only Moscow and Shanghai (China and Russia–both UMICs), and Pyongyang (DPRK, a Low-Income Country), host WHOCCs in LMICs. While WHOCCs in high-income countries have much to offer LMICs, the voices of the many South based institutions already recognized for their expertise in PHC need to be heard.

We recognise the limitations of this study. The data for analysis of the WHOCCs on PHC is limited to publicly available data from the WHO website. It reflects their intended subjects, objectives and activities at designation, rather than the implementation of those activities or outcomes. Our analysis focus on WHOCCs and whether their presence/absence/focus on PHC is one of the indicators of WHO priority in strengthening health systems in the region and countries. Relevant support on PHC to WHO is not limited to those WHOCCs with a specific designation on PHC: WHOCCs on UHC, on health systems strengthening and related areas will offer valuable inputs. Future studies can be conducted to validate the findings of the publicly available data presented in this paper. There is insufficient information in the public domain to confidently account for the limited distribution of collaborating centers on PHC in LMICs, with further dialogue and research recommended in this area.

## Conclusions

"Primary Health Care is essential health care based on practical, scientifically sound and socially acceptable methods and technology made universally accessible to individuals and families in the community through their full participation and at a cost that the community and country can afford to maintain at every stage of their development in the spirit of self-

reliance and self-determination" [1]. That incisive 1978 Alma-Ata vision has been adopted by successive waves of health and development policy.

The challenge to provide geographic and financial access to health care for whole populations has been reiterated in the universality of UHC, and its commitment to financial protection [14]. The socially acceptable essential health care and technology has been revisited in service reforms [6] and rearticulated in people-centred primary care and family medicine [7]. The multisectoral engagement of PHC has been reimagined in the social determinants of health [5], reframed in "health in all policies" [15, 16], embodied in the comprehensiveness of the SDGs [17]. They are all restated in the Astana Declaration. Achieving health and wellbeing for all at all ages needs the expanded vision of PHC if it is to address the social determinants, to ensure equity and access, to integrate health and wellbeing into sustainable development. But there is a risk if PHC merely devolves to Primary Care, if the doctor at the centre of Family Medicine displaces the community engagement that envelopes each individual in holistic, people-centred health care.

The next wave of designations for WHOCCs on PHC needs representation from all regions, all economies, particularly where representation is currently absent or limited. The WHOCC Ghent is currently building on its Primafamed Family Medicine network [18] to promote African institutions as WHOCC candidates on PHC, but there is need to complement this with Southern institutions based in Asia and South-East Asia, in the Pacific, and the Americas. The next wave needs a network that promotes South-South collaboration and builds on lessons learned in the implementation of PHC across diverse health systems. It needs to invest in the health systems transitions being resourced by sustainable development, providing WHO with feedback on policy and programme implementation at country level.

If PHC was the key to health for all in 2000 [1] and needed 'now more than ever' in 2008 [6], that urgency remains current. Post Astana, the network of WHOCCs on PHC needs to support WHO in reframing the positioning of PHC in the global discourses it has initiated, and the global policies that share its vision and values.

## Author Contributions

**Conceptualization:** Resham B. Khatri, Peter S. Hill, Yibeltal Assefa.

**Data curation:** Resham B. Khatri, Eskinder Wolka, Frehiwot Nigatu, Anteneh Zewdie.

**Formal analysis:** Resham B. Khatri, Peter S. Hill, Yibeltal Assefa.

**Investigation:** Resham B. Khatri, Frehiwot Nigatu, Yibeltal Assefa.

**Methodology:** Resham B. Khatri, Peter S. Hill, Anteneh Zewdie, Yibeltal Assefa.

**Project administration:** Resham B. Khatri, Eskinder Wolka, Frehiwot Nigatu, Anteneh Zewdie, Yibeltal Assefa.

**Resources:** Resham B. Khatri, Eskinder Wolka, Frehiwot Nigatu, Yibeltal Assefa.

**Software:** Resham B. Khatri.

**Supervision:** Peter S. Hill, Yibeltal Assefa.

**Validation:** Resham B. Khatri, Peter S. Hill, Eskinder Wolka, Frehiwot Nigatu, Anteneh Zewdie, Yibeltal Assefa.

**Visualization:** Resham B. Khatri, Peter S. Hill, Eskinder Wolka, Frehiwot Nigatu, Anteneh Zewdie, Yibeltal Assefa.

**Writing – original draft:** Resham B. Khatri, Peter S. Hill, Yibeltal Assefa.

**Writing – review & editing:** Resham B. Khatri, Peter S. Hill, Eskinder Wolka, Frehiwot
Nigatu, Anteneh Zewdie, Yibeltal Assefa.

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
