## [Decision Letter · Decision Letter 0]

31 May 2023

PGPH-D-23-00280

Beyond Astana: Configuring the World Health Organization Collaborating Centres for Primary Health Care

Dear Dr. Khatri,

Thank you for submitting your manuscript to PLOS Global Public Health. After careful consideration, we feel that it has merit but does not fully meet PLOS Global Public Health’s publication criteria as it currently stands. Therefore, we invite you to submit a revised version of the manuscript that addresses the points raised during the review process.

We look forward to receiving your revised manuscript.

Kind regards,

Shifa S. Habib

Academic Editor

Journal Requirements:

Additional Editor Comments (if provided):

Reviewers' comments:

Reviewer's Responses to Questions

**Comments to the Author**

1. Does this manuscript meet PLOS Global Public Health’s publication criteria? Is the manuscript technically sound, and do the data support the conclusions? The manuscript must describe methodologically and ethically rigorous research with conclusions that are appropriately drawn based on the data presented.

Reviewer #1: Partly

Reviewer #2: Yes

2. Has the statistical analysis been performed appropriately and rigorously?

Reviewer #1: N/A

Reviewer #2: N/A

3. Have the authors made all data underlying the findings in their manuscript fully available (please refer to the Data Availability Statement at the start of the manuscript PDF file)?

Reviewer #1: Yes

Reviewer #2: Yes

4. Is the manuscript presented in an intelligible fashion and written in standard English?

Reviewer #1: Yes

Reviewer #2: Yes

5. Review Comments to the Author

Reviewer #1: Although the manuscript sums up the revolution of PHC but a more interesting aspect would be to discuss why LMICs are still lacking. It will be amazing if the authors can also summarize the revolution in terms of flow charts so its easier to follow.Good luck with the paper.

Reviewer #2: The paper is interesting, informative, and highlighting the important issue of WCCs absence from the African region and from LMICs. It would have been interesting if some impact of these centres on the improvement of the health systems or on improving in the number/ support to the country's work force or on capacity development would have been very useful. Authors have mentioned this and absence of information on implementation as their limitation. it is still an important contribution. My few minor comments are as follows; declaration of interest is missing, on page 8 first paragraph please make corrections in Russia and China being mentioned as LMIC, also this does not match with he information given in the Table-1. make same correction in the discussion section as well. On page 16 last paragraph please correct the font size to match rest of the document

6. PLOS authors have the option to publish the peer review history of their article (what does this mean?). If published, this will include your full peer review and any attached files.

**Do you want your identity to be public for this peer review?** For information about this choice, including consent withdrawal, please see our Privacy Policy.

Reviewer #1: No

Reviewer #2: **Yes: **Sarah Saleem

---

## [Editor Report · Decision Letter 1]

6 Jul 2023

Beyond Astana: Configuring the World Health Organization Collaborating Centres for Primary Health Care

PGPH-D-23-00280R1

Dear Dr. Khatri,

We are pleased to inform you that your manuscript 'Beyond Astana: Configuring the World Health Organization Collaborating Centres for Primary Health Care' has been provisionally accepted for publication in PLOS Global Public Health.

Best regards,

Shifa S. Habib

Academic Editor